# Systematic Review of Large Language Models: Applications, Limitations, Practical Usages and Future Directions

## Abstract

Large Language Models have revolutionized natural language processing with their remarkable ability to understand and generate human-like text. This review explores the various applications of large language models, highlighting their versatility across different domains. The paper begins with an introduction to LLMs, followed by an overview of their types and a detailed literature review. We then examine their limitations before delving into specific applications such as text generation, translation, summarization, and more. Finally, we discuss future directions for research and development, concluding with a summary of key findings and the potential impact of large language models on various industries.

## 1 Introduction

Large Language Models (LLMs) represent a significant breakthrough in the field of artificial intelligence (AI), particularly in natural language processing (NLP) Chang et al. (2024); Zhao et al. (2023); Thirunavukarasu et al. (2023). These models are designed to understand, interpret, and generate human language with unprecedented accuracy and coherence. The development of LLMs, such as OpenAI's GPT series Achiam et al. (2023) and Google's BERT Devlin (2018), has been propelled by advancements in deep learning and the availability of vast computational resources. These models have the capacity to analyze and generate text based on extensive training datasets, enabling them to perform a wide range of language-related tasks that were previously beyond the reach of AI.

The evolution of LLMs can be traced back to the introduction of the Transformer architecture by Vaswani (2017), which replaced the traditional recurrent, deep neural networks and long short-term memory (LSTM) networks with self-attention mechanisms Li et al. (2020). This innovation allowed for parallel processing of data and improved the scalability of models, laying the groundwork for subsequent developments in LLMs. The Transformer architecture's ability to capture contextual relationships in text has been fundamental to the success of models like BERT and Chat GPT, which have set new benchmarks in NLP tasks.

LLMs have demonstrated remarkable capabilities in various applications, from language translation Zhang et al. (2023a) and text summarization Zhang et al. (2024) to sentiment analysis Zhang et al. (2023b); Deng et al. (2023) and question answering Ko et al. (2023); **?**. For instance, GPT-3's ability to generate human-like text has been leveraged in creative writing, customer service, and even coding. BERT, on the other hand, excels in understanding context and meaning, making it highly effective for tasks such as question answering and sentiment analysis. These applications highlight the versatility of LLMs and their potential to transform numerous industries by automating and enhancing complex language tasks.

However, the deployment of LLMs usually has some challenges. One of the primary concerns is the resource intensity required for training and deploying these models. The vast computational power and memory needed to handle models with billions of parameters can be prohibitive, particularly for smaller organizations. Additionally, LLMs can perpetuate and even amplify biases present in their training data, leading to ethical concerns regarding fairness and discrimination. Addressing these issues is crucial to ensure that LLMs are used responsibly and equitably across different applications.

Looking ahead, the future of LLMs lies in overcoming these limitations and expanding their capabilities. Research is ongoing to develop more efficient training methods, reduce bias, and improve the interpretability of these models. Techniques such as model pruning, knowledge distillation, and fairness-aware training algorithms are being explored to make LLMs more accessible and reliable. Furthermore, integrating LLMs with other data types, such as images and audio, could unlock new multimodal applications, further broadening the scope of what these powerful models can achieve. As LLMs continue to evolve, they hold the promise of driving innovation and improving human-computer interactions in unprecedented ways.

## 2   TYPES OF LLMS

LLMs can be broadly categorized based on their architecture and training objectives:

Generative Models: These models, such as GPT-3 and GPT-4, are designed to generate text based on a given prompt Floridi (2023); ?); Kalyan (2023). They excel in creative writing, conversation generation, and other text generation tasks. Generative models typically use a decoder-only transformer architecture, focusing on generating the next word in a sequence given the preceding context.

Masked Language Models: BERT (Bidirectional Encoder Representations from Transformers) Devlin (2018); Kenton & Toutanova (2019) and its variants fall into this category. They are trained to predict missing words in a sentence, making them highly effective for understanding context and meaning. Masked language models use an encoder-only architecture, enabling them to consider both the left and right context of a word simultaneously.

Sequence-to-Sequence Models: These models, including T5 (Text-to-Text Transfer Transformer) and BART (Bidirectional and Auto-Regressive Transformers) Ozdemir (2023), are designed for tasks that involve transforming one sequence into another, such as translation and summarization. Sequence-to-sequence models typically use an encoder-decoder architecture, where the encoder processes the input sequence and the decoder generates the output sequence.

Hybrid Models: Models like XLNet Yang (2019) combine features from both generative and masked models to improve performance on a wider range of tasks. XLNet, for example, uses a permutation-based training objective that allows it to capture bidirectional context like BERT while retaining the autoregressive properties of GPT.

These different types of LLMs reflect the diverse approaches to leveraging the transformer architecture for various NLP tasks. Each type has its strengths and is suited to specific applications, contributing to the versatility of LLMs in handling a wide range of language-related challenges.

## 3   LITERATURE REVIEW

Research on large language models (LLMs) has seen exponential growth over the past decade, focusing on enhancing their capabilities and exploring their applications across various domains. This section reviews significant studies and developments in the field, highlighting how LLMs are employed for different tasks.

### 3.1   DEEP LEARNING METHODS AND TECHNIQUES USED TO DEVELOP LLMS

Deep Learning (DL) methods and techniques used to develop large language models (LLMs) include:

- **Transformer Architecture**: The backbone of most modern LLMs, such as BERT, GPT, and T5, which uses self-attention mechanisms to handle long-range dependencies in text data.
- **Self-Attention Mechanisms**: Allow the model to weigh the importance of different words in a sentence when making predictions, enhancing the model's ability to understand context and meaning.
- **Transfer Learning**: Pre-training models on large datasets and then fine-tuning them on specific tasks to improve performance and reduce the need for extensive labeled data.

- **Bidirectional Training**: Used in models like BERT, where the context of a word is learned from both its preceding and following words, leading to better language understanding.
- **Generative Pre-training**: Used in GPT models where the model is trained to generate text by predicting the next word in a sequence, allowing it to learn language patterns and structure.
- **Sequence-to-Sequence Learning**: Employed in models like T5 and BART, which transform input sequences into output sequences, making them suitable for tasks like translation and summarization.
- **Denoising Autoencoders**: Used in models like BART to reconstruct original data from corrupted input, helping the model to learn robust representations.
- **Few-Shot Learning**: Techniques that allow models like GPT-3 to perform tasks with very few examples, demonstrating the ability to generalize from limited data.
- **Parallel Processing**: Leveraging GPU and TPU hardware to process multiple parts of the input simultaneously, significantly speeding up training times and enabling the development of very large models.
- **Masked Language Modeling**: A training technique used in BERT where random words in a sentence are masked and the model learns to predict them, encouraging the model to build a deeper understanding of the language context.

## 3.2 EARLY DEVELOPMENT AND CORE ARCHITECTURES

The foundation of modern LLMs lies in the introduction of the Transformer architecture by Vaswani (2017). This architecture replaced traditional RNNs and LSTMs with self-attention mechanisms, enabling parallel processing and significantly improving scalability. The Transformer architecture's ability to capture long-range dependencies in text has been fundamental to the success of subsequent LLMs like BERT and GPT, which have achieved state-of-the-art performance in numerous NLP benchmarks.

## 3.3 LANGUAGE UNDERSTANDING AND CONTEXTUAL EMBEDDINGS

One of the pivotal advancements in LLM research was the development of BERT (Bidirectional Encoder Representations from Transformers) by Devlin (2018). BERT's innovative use of bidirectional training for language modeling allowed it to understand the context of a word based on both its preceding and following words. This bidirectional approach enabled BERT to excel in various tasks, including named entity recognition, question answering, and sentiment analysis. Studies have shown that BERT's contextual embeddings significantly improve performance in these tasks compared to previous models.

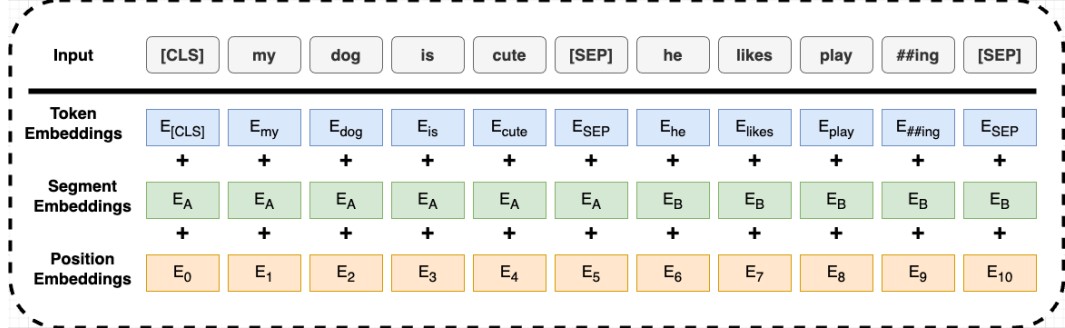

Figure 1: BERT input representation. The input embeddings are the sum of the token embeddings, the segmentation embeddings and the position embeddings.

## 3.4 GENERATIVE MODELS AND TEXT GENERATION

Generative Pre-trained Transformer models, particularly GPT-2 and GPT-3 by OpenAI, have demonstrated exceptional capabilities in text generation. The authors in Radford et al. (2019) showcased

GPT-2's ability to generate coherent and contextually relevant text by training on diverse internet text. GPT-3, with its 175 billion parameters, took this further by exhibiting strong performance in few-shot, one-shot, and zero-shot learning settings Mann et al. (2020). These models have been applied to various creative tasks, including writing essays, poetry, and generating code snippets, highlighting their versatility.

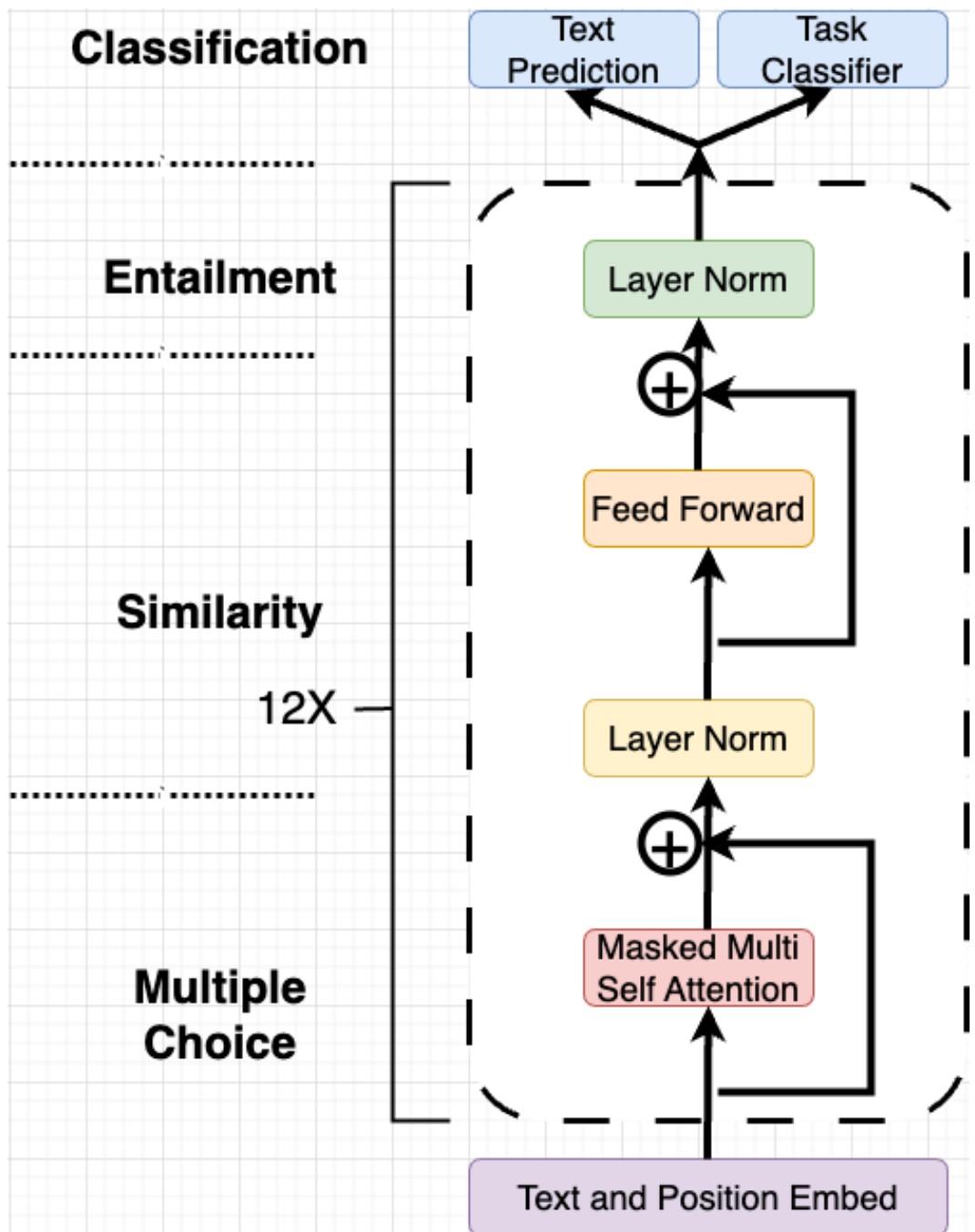

Figure 2: Transformer architecture and training objectives.

## 3.5 SEQUENCE-TO-SEQUENCE MODELS FOR TRANSLATION AND SUMMARIZATION

Sequence-to-sequence models like T5 (Text-to-Text Transfer Transformer) and BART (Bidirectional and Auto-Regressive Transformers) have been instrumental in tasks that involve transforming one sequence of text into another, such as translation and summarization. The authors in Raffel et al.

(2020) introduced T5, a model that frames all NLP tasks as text-to-text problems, leading to robust performance across multiple benchmarks. The authors in Liu (2020) presented BART, which combines bidirectional and autoregressive transformers, proving effective in denoising autoencoding tasks and achieving state-of-the-art results in text summarization.

### 3.6 FINE-TUNING AND TRANSFER LEARNING

The adaptability of LLMs through fine-tuning has been a major focus of research. The author in Houlsby et al. (2019) introduced Adapter modules, which allow efficient fine-tuning by adding small, task-specific modules to pre-trained models. This approach reduces the computational resources required for fine-tuning, making it feasible to apply LLMs to a wider range of tasks and domains. Fine-tuning techniques have enabled models like BERT and GPT to be customized for specific applications, improving their performance and utility in various contexts.

### 3.7 ADDRESSING BIAS AND FAIRNESS

LLMs have been used for perpetuating biases present in their training data. The author in Bender et al. (2021) highlighted the ethical implications of large-scale language models, emphasizing the need for fairness-aware algorithms and bias mitigation strategies. Research in this area has focused on developing methods to detect and reduce biases in LLM outputs, ensuring that these models can be used responsibly and equitably.

### 3.8 MULTIMODAL APPLICATIONS

Recent studies have explored the integration of LLMs with other modalities, such as images and audio. For example, models like CLIP (Contrastive Language–Image Pre-training) Fang et al. (2022) by OpenAI combine text and image data to improve understanding and generation capabilities across modalities. This multimodal approach opens up new possibilities for applications in areas like augmented reality, multimedia content creation, and more.

### 3.9 RECENT DEVELOPMENTS AND BENCHMARKS

Recent years have seen the emergence of even larger and more capable LLMs, along with new benchmarks to evaluate their performance. Models such as GPT-4 and PaLM have pushed the boundaries of what is possible with LLMs. GPT-4, for example, incorporates advanced reasoning capabilities and can handle more complex tasks with higher accuracy. PaLM (Pathways Language Model) leverages a more efficient training paradigm to achieve state-of-the-art results across various tasks.

#### 3.9.1 HELM BENCHMARK

The Holistic Evaluation of Language Models (HELM) benchmark provides a comprehensive framework for assessing LLM performance across a wide range of tasks, including language understanding, generation, and reasoning. By incorporating diverse metrics and scenarios, HELM offers a nuanced understanding of model capabilities and limitations.

#### 3.9.2 LMSYS CHATBOT ARENA LEADERBOARD

The LMSYS Chatbot Arena Leaderboard evaluates the conversational abilities of different chatbots, offering insights into their performance in real-world dialogue settings. This benchmark focuses on aspects such as coherence, relevance, and user satisfaction, providing a practical measure of how well LLMs perform in interactive applications.

### 3.10 HALLUCINATION IN LARGE LANGUAGE MODELS

A critical aspect of LLMs is the phenomenon of hallucination, where models generate information that is not present in the input data or factual knowledge base. This issue has been highlighted in recent research as a significant drawback of current LLMs. Hallucinations can lead to the dissemination of misinformation and reduce trust in AI systems. Various strategies, such as improved training

methods, better model architectures, and post-processing techniques, are being explored to mitigate this issue Ji et al. (2023).

### 3.11 COMPARISON WITH RECENT REVIEWS

To provide a more comprehensive overview, we compare our findings with recent reviews on LLMs. Notably, Bommasani et al. (2021) and Zhao et al. (2023) offer extensive analyses of the latest advancements and applications of LLMs, including ethical considerations and deployment challenges. These reviews highlight the importance of continuous benchmarking and evaluation to ensure that LLMs are developed and used responsibly.

By integrating insights from recent benchmarks and reviews, this section provides a broader perspective on the current state of LLM research, highlighting both the progress made and the challenges that remain.

Table 1: Comparative Analysis of LLMs

| Model | Pros | Cons | Datasets | Metrics |
|-------|------|------|----------|---------|
| BERT | Bidirectional context, strong NLP performance | Large model size, inference cost | Wikipedia, BooksCorpus | Accuracy, F1 score |
| GPT-3 | Few-shot learning, versatile generation | High computational cost, hallucination risk | Diverse internet text | Perplexity, BLEU |
| T5 | Unified text-to-text framework, flexibility | High training cost, complex architecture | C4 dataset | ROUGE, BLEU |
| BART | Effective in summarization, denoising | Large memory requirements, slower training | XSUM, CNN/DailyMail | ROUGE, BLEU |

## 4 COMPARATIVE ANALYSIS OF LLMs

Comparing different LLMs, such as GPT, BERT, and T5, reveals distinct strengths and applications tailored to specific tasks. BERT (Bidirectional Encoder Representations from Transformers) is designed for tasks requiring a deep understanding of context, excelling in question answering and sentiment analysis due to its bidirectional training approach. BERT's ability to consider context from both directions makes it particularly effective for tasks where understanding the nuance of language is critical. GPT (Generative Pre-trained Transformer), particularly in its latest iterations like GPT-3, is renowned for its generative capabilities, producing coherent and contextually relevant text, making it ideal for creative writing and content generation. GPT-3's ability to generate human-like text has been utilized in various applications, from chatbots to content creation tools. T5 (Text-to-Text Transfer Transformer) frames all NLP tasks as text-to-text problems, offering a unified approach that simplifies task adaptation and improves performance across a variety of benchmarks. T5's flexibility allows it to be easily fine-tuned for different tasks, making it a versatile tool in the NLP toolkit. Analyzing these models' architectures, training methodologies, and performance metrics provides insights into their suitability for different applications and helps guide their optimal use in various contexts Raffel et al. (2020).

Recent studies have conducted comprehensive analyses of various LLMs, examining their strengths and weaknesses, the datasets used, evaluation metrics, and overall performance. For instance, Bommasani et al. (2021) and Zhao et al. (2023) provide detailed comparisons of models like BERT, GPT-3, T5, and newer architectures. These reviews highlight the trade-offs in model complexity, training efficiency, and performance on different NLP tasks.

## 5 Adversarial Robustness

LLMs are susceptible to adversarial attacks, where malicious inputs are crafted to deceive the model into producing incorrect or harmful outputs. Research in adversarial robustness aims to develop methods to detect and defend against such attacks. Techniques such as adversarial training, where models are exposed to adversarial examples during training, and robust optimization strategies are being explored to enhance model resilience. Ensuring the robustness of LLMs is critical for their deployment in sensitive applications, such as healthcare and finance, where reliability and accuracy are paramount. For example, in medical applications, ensuring that LLMs are robust against adversarial inputs can prevent incorrect diagnoses and treatment recommendations. Similarly, in finance, robust models can protect against fraud attempts that exploit model weaknesses. By enhancing the adversarial robustness of LLMs, their trustworthiness and reliability in critical applications are significantly improved, paving the way for broader adoption in high-stakes environments Mann et al. (2020).

## 6 Limitations of LLMs

Despite their impressive capabilities, LLMs have several limitations that need to be addressed:

Bias and Fairness: LLMs can perpetuate and even amplify biases present in their training data, leading to unfair and potentially harmful outcomes. For example, biases related to gender, race, and socioeconomic status can manifest in generated text, affecting applications in sensitive areas such as hiring or law enforcement. Ensuring fairness and mitigating bias requires ongoing research and robust ethical guidelines Gallegos et al. (2024); Li et al. (2023).

Resource Intensity: Training and deploying LLMs require significant computational resources, including large amounts of memory and processing power. This resource intensity makes it challenging for smaller organizations to access and utilize these models, potentially leading to a concentration of AI capabilities in the hands of a few large entities Bai et al. (2024).

Interpretability: LLMs operate as black boxes, making it difficult to understand how they arrive at specific decisions or predictions. This lack of interpretability can hinder trust and acceptance, particularly in high-stakes applications such as healthcare and legal systems. Developing methods to explain and interpret the outputs of LLMs is a critical area of research Zhao et al. (2024); Singh et al. (2023).

Overfitting and Generalization: While LLMs can perform well on a wide range of tasks, they can also overfit to specific datasets and struggle to generalize to unseen data or tasks. Ensuring that these models can adapt and perform robustly across diverse contexts remains a challenge Anil et al. (2022); Chang et al. (2024); Tirumala et al. (2022).

## 7 Ethical Considerations

The deployment of Large Language Models (LLMs) presents significant ethical considerations that must be addressed to ensure their responsible use. One primary concern is the potential for misuse, such as generating misleading or harmful content Ong et al. (2024); Watkins (2023); Meyer et al. (2023). LLMs can produce text that is indistinguishable from human-written content, making them powerful tools for spreading misinformation, creating fake news, or even engaging in social engineering attacks. Additionally, privacy concerns arise when LLMs are trained on large datasets that may include sensitive or personal information. Ensuring that training data is anonymized and secure is critical to protecting individuals' privacy Solomon & Woubie (2024). Furthermore, the societal impact of automated content generation, including job displacement in sectors reliant on human creativity and communication, must be carefully considered. Ethical guidelines and regulatory frameworks are essential to navigate these challenges, promoting the beneficial use of LLMs while mitigating risks. Moreover, addressing biases within these models is crucial as they often reflect societal prejudices present in the training data, which can perpetuate stereotypes and lead to discriminatory practices. These ethical dimensions highlight the need for ongoing vigilance and proactive measures to ensure that LLMs are developed and used in ways that are fair, transparent, and aligned with societal values Bender et al. (2021).

## 8   EVALUATION METRICS AND BENCHMARKS

Evaluating LLMs involves a variety of metrics and benchmarks tailored to specific tasks. Common metrics include BLEU (Bilingual Evaluation Understudy) Papineni et al. (2002) scores for machine translation, which measure the accuracy of translated text against reference translations. F1 scores Chicco & Jurman (2020), which consider both precision and recall, are used for classification tasks to evaluate a model's accuracy. Perplexity is another metric used to assess language models, indicating how well a model predicts a sample. Benchmark datasets such as GLUE (General Language Understanding Evaluation) Wang (2018) and SQuAD (Stanford Question Answering Dataset) Rajpurkar (2016); Rajpurkar et al. (2018) provide standardized tests for assessing LLM performance across multiple tasks. These metrics and benchmarks are essential for comparing different models, identifying strengths and weaknesses, and guiding future improvements. Additionally, real-world performance tests, where models are deployed in practical scenarios, offer valuable insights into their effectiveness and reliability. This comprehensive evaluation approach ensures that LLMs are robust, reliable, and ready for deployment in diverse applications Devlin (2018).

## 9   FUTURE DIRECTIONS

Looking ahead, the future of LLMs involves addressing current limitations, such as resource intensity and interpretability, while expanding their capabilities through continual learning and integration with other data types. Techniques like model pruning and knowledge distillation are being researched to reduce the computational footprint of LLMs, making them more accessible. Improving interpretability through attention visualization and post-hoc explanation methods is also a key area of focus, aimed at building trust and transparency in AI systems. The future of LLMs lies in addressing current limitations and expanding their capabilities:

Ethical and Fair AI: Ensuring that LLMs do not perpetuate biases present in training data is crucial. Research is focusing on developing techniques to mitigate these biases Bender et al. (2021). Approaches such as bias detection, fairness-aware training algorithms, and post-processing techniques aim to create more equitable AI systems.

Efficiency and Accessibility: Reducing the computational resources required to train and deploy LLMs will make them more accessible. Techniques like model pruning and knowledge distillation are promising in this regard. These methods aim to reduce model size and inference time without significantly compromising performance, making LLMs more practical for deployment in resource-constrained environments.

Multimodal Models: Integrating LLMs with other types of data, such as images and audio, can create more comprehensive AI systems capable of understanding and generating content across different media. Multimodal models can enhance applications such as video captioning, audio-visual scene understanding, and cross-modal retrieval, enabling richer and more interactive user experiences.

Human-AI Collaboration: Enhancing the ability of LLMs to work alongside humans in creative and analytical tasks can lead to more productive and innovative outcomes. Research is exploring ways.

## 10   CONCLUSION

In summary, large language models (LLMs) have significantly advanced the field of natural language processing, enabling machines to understand and generate human language with remarkable accuracy. Their applications span a wide range of domains, including text generation, translation, summarization, sentiment analysis, and question answering. By leveraging vast amounts of training data and sophisticated architectures like transformers, LLMs have set new benchmarks for various NLP tasks. Their ability to handle diverse and complex language-related challenges underscores their transformative potential across different industries.

Despite their impressive capabilities, LLMs face several limitations that must be addressed to fully realize their potential. Issues such as bias and fairness, resource intensity, and the need for greater interpretability pose significant challenges. Ensuring that LLMs are used ethically and responsibly is crucial to prevent misuse and to foster trust in their applications. Research efforts are increasingly

focusing on developing techniques to mitigate these issues, such as fairness-aware algorithms, model efficiency improvements, and methods for enhancing interpretability.

Looking forward, the future of LLMs is promising yet demands continuous innovation and vigilance. As researchers and practitioners work to overcome current limitations, the integration of LLMs with multimodal data and advancements in continual learning will likely expand their applicability and effectiveness. Ensuring equitable access to these powerful models will be vital to democratize their benefits. As LLMs evolve, they have the potential to drive significant progress in AI, improving human-computer interactions and contributing to advancements in various fields, from healthcare to education and beyond. The journey of LLMs is just beginning, and their continued development will shape the future of technology and its impact on society.

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
