# OpenReview forum: "Systematic Review of Large Language Models: Applications, Limitations, Practical Usages and Future Directions"
_ICLR.cc/2025/Conference — Submitted to ICLR 2025_

### Official Review · Reviewer_4Db8 · 2024-11-02

**Soundness:** 1
**Presentation:** 1
**Contribution:** 1
**Rating:** 1
**Confidence:** 5

**Summary:**

This paper provides a brief summary and review of LLM architecture, and its limitations, as well as various application areas (e.g., text generation, translation, summarization, etc), and existing LLM benchmarks. They also discuss future directions for LLM research.

**Strengths:**

1. Provides a summary of LLM architecture, application areas, limitations, practical usage, and future directions. Which could be useful to get a quick idea for new LLM researchers to get the basic idea. However, it is more effective as a blog post and not at all a research survey paper.

**Weaknesses:**

i. This survey does not provide any new novel insights in comparison to what is known already about LLMs (e.g., [1], [2] etc.). It is more of a straightforward summary of the architectures, applications, and limitations of LLMs, lacking in-depth critical review.

ii. Moreover, although the paper title claims the paper as a systematic survey, the discussion on different topics in this review is also superficial.

iii. Even though the paper mentions LLMs, the discussion is more around typical transformer-based language models like BERT, GPT, and T5 without offering new insights.

iv. The paper relies heavily on vague citations. Moreover, some of the citations have "?" marks. This demonstrates that the paper lacks attention to detail. Potentially, this paper was written without any comprehensive research.

1. Minaee, S., Mikolov, T., Nikzad, N., Chenaghlu, M., Socher, R., Amatriain, X. and Gao, J., 2024. Large language models: A survey. arXiv preprint arXiv:2402.06196.

2. Zhao, W.X., Zhou, K., Li, J., Tang, T., Wang, X., Hou, Y., Min, Y., Zhang, B., Zhang, J., Dong, Z. and Du, Y., 2023. A survey of large language models. arXiv preprint arXiv:2303.18223.

v. The discussed limitations and proposed future directions also do not offer anything new.

vi. Figure 2 also looks pretty bad. The text size in the caption is also very large in comparison to the paper text.

**Questions:**

See above

---

### Official Review · Reviewer_kJ2V · 2024-11-03

**Soundness:** 1
**Presentation:** 1
**Contribution:** 1
**Rating:** 1
**Confidence:** 5

**Summary:**

This is a review paper on a broad topic of LLMs regarding the types, applications, and limitations, etc., of LLMs.

**Strengths:**

It is important to provide a review on LLMs in the era of the vast development of LLMs.

**Weaknesses:**

I believe this paper is not appropriate for ICLR. This is not a "systematic" review. The content is superficial and outdated. The insights are not valid.

**Questions:**

Dear author, I think it is more realistic to focus on a certain smaller aspect and conduct a really "systematic" review on that topic. The current paper is not a good review, as the topic is very big and you didn't properly address this field in such a small paper. I would recommend to rethink the scope.

---

### Official Review · Reviewer_1rVe · 2024-11-04

**Soundness:** 1
**Presentation:** 1
**Contribution:** 1
**Rating:** 1
**Confidence:** 4

**Summary:**

This paper presents a literature review on the topic of Large Language Models. The paper characterizes different types of LLMs: Generative Models, Masked Language Models, Sequence-to-Sequence Models, and, Hybrid Models. The survey discusses a range of topics, from "Deep-learning methods and techniques used to develop LLMs" to "Recent developments and Benchmarks".

**Strengths:**

None, as can be seen in the manuscript, Section 3.11 (Comparison with Recent Reviews):
>To provide a more comprehensive overview, we compare our findings with recent reviews on LLMs. Notably, Bommasani et al. (2021) and Zhao et al. (2023) offer extensive analyses of the latest advancements and applications of LLMs, including ethical considerations and deployment challenges. These reviews highlight the importance of continuous benchmarking and evaluation to ensure that
LLMs are developed and used responsibly. **By integrating insights from recent benchmarks and reviews, this section provides a broader perspective on the current state of LLM research, highlighting both the progress made and the challenges that remain.**

That is, this paper does not offer any contribution other than those mentioned in Bommasani et al. (2021) and Zhao et al. (2023).

**Weaknesses:**

* The paper does not present any novel or meaningful contribution.
* The content is outdated.
* The organization, particularly in Section 3 is very poor.
* There are missing or wrongly cited references.
* **The paper feels empty:** Most of the subsections in Section 3 contain a single paragraph with (in the best case) a single reference.

**Questions:**

None

---

### Official Review · Reviewer_Zppz · 2024-11-06

**Soundness:** 1
**Presentation:** 1
**Contribution:** 1
**Rating:** 1
**Confidence:** 4

**Summary:**

This paper provides an overview of the development, applications, and comparative analysis of Language Models (LMs). It begins by detailing the methodologies used in the construction of LMs. Following this, the paper explores various applications of LMs, and finally, the study concludes with a side-by-side comparison of four LMs, evaluating their strengths and weaknesses.

**Strengths:**

The paper provides an overview of the technologies employed in the development of Language Models (LMs).

**Weaknesses:**

The paper’s objectives are not clearly defined. While it purports to review Large Language Models (LLMs), the models it examines are not currently regarded as large by contemporary standards (such as open models like LLaMa-2, OLMo, etc.) [1]. Furthermore, despite claiming to explore future directions for LLMs, the paper fails to address this topic adequately. For instance, specific future applications of LLMs in new fields or emerging challenges associated with the expansion of LLM could have been explored [2].

[1] Bommasani, Rishi, et al. "On the opportunities and risks of foundation models." arXiv preprint arXiv:2108.07258 (2021).

[2] Li, Sha, et al. "Defining a new NLP playground." arXiv preprint arXiv:2310.20633 (2023).

**Questions:**

I have no questions.

---

### Meta-Review · Area_Chair_WG7o · 2024-12-20

**Metareview:**

This paper attempts to provide an overview of large language modes including architecture, data, training and algorithms. No strengths were highlighted by the reviewers. All reviewers agree that this paper is poorly written, does not offer new insights not previously known, contains outdated information, and contains many typos, missing citations, figures. Reviewers have also raised concerns that it potentially might be LM-generated, and I share that concern.

**Additional Comments On Reviewer Discussion:**

The authors provided no rebuttal.

---

### Decision · Program_Chairs · 2025-01-22

Reject